# Fast learning rates with heavy-tailed losses

**Vu Dinh**[1]   **Lam Si Tung Ho**[2]   **Duy Nguyen**[3]   **Binh T. Nguyen**[4]

[1]Program in Computational Biology, Fred Hutchinson Cancer Research Center
[2]Department of Biostatistics, University of California, Los Angeles
[3]Department of Statistics, University of Wisconsin-Madison
[4]Department of Computer Science, University of Science, Vietnam

## Abstract

We study fast learning rates when the losses are not necessarily bounded and may have a distribution with heavy tails. To enable such analyses, we introduce two new conditions: (i) the envelope function $\sup_{f \in \mathcal{F}} |\ell \circ f|$, where $\ell$ is the loss function and $\mathcal{F}$ is the hypothesis class, exists and is $L^r$-integrable, and (ii) $\ell$ satisfies the multi-scale Bernstein's condition on $\mathcal{F}$. Under these assumptions, we prove that learning rate faster than $O(n^{-1/2})$ can be obtained and, depending on $r$ and the multi-scale Bernstein's powers, can be arbitrarily close to $O(n^{-1})$. We then verify these assumptions and derive fast learning rates for the problem of vector quantization by $k$-means clustering with heavy-tailed distributions. The analyses enable us to obtain novel learning rates that extend and complement existing results in the literature from both theoretical and practical viewpoints.

## 1 Introduction

The rate with which a learning algorithm converges as more data comes in play a central role in machine learning. Recent progress has refined our theoretical understanding about setting under which fast learning rates are possible, leading to the development of robust algorithms that can automatically adapt to data with hidden structures and achieve faster rates whenever possible. The literature, however, has mainly focused on bounded losses and little has been known about rates of learning in the unbounded cases, especially in cases when the distribution of the loss has heavy tails [van Erven et al., 2015].

Most of previous work about learning rate for unbounded losses are done in the context of density estimation [van Erven et al., 2015, Zhang, 2006a,b], of which the proofs of fast rates implicitly employ the central condition [Grünwald, 2012] and cannot be extended to address losses with polynomial tails [van Erven et al.]. Efforts to resolve this issue include Brownlees et al. [2015], which proposes using some robust mean estimators to replace empirical means, and Cortes et al. [2013], which derives relative deviation and generalization bounds for unbounded losses with the assumption that $L^r$-diameter of the hypothesis class is bounded. However, results about fast learning rates were not obtained in both approaches. Fast learning rates are derived in Lecué and Mendelson [2013] for sub-Gaussian losses and in Lecué and Mendelson [2012] for hypothesis classes that have sub-exponential envelope functions. To the best of our knowledge, no previous work about fast learning rates for heavy-tailed losses has been done in the literature.

The goal of this research is to study fast learning rates for the empirical risk minimizer when the losses are not necessarily bounded and may have a distribution with heavy tails. We recall that heavy-tailed distributions are probability distributions whose tails are not exponentially bounded: that is, they have heavier tails than the exponential distribution. To enable the analyses of fast rates with heavy-tailed losses, two new assumptions are introduced. First, we assume the existence and the $L^r$-integrability of the *envelope function* $F = \sup_{f \in \mathcal{F}} |f|$ of the hypothesis class $\mathcal{F}$ for

some value of $r \geq 2$, which enables us to use the results of Lederer and van de Geer [2014] on concentration inequalities for suprema of empirical unbounded processes. Second, we assume that the loss function satisfies the *multi-scale Bernstein's condition*, a generalization of the standard Bernstein's condition for unbounded losses, which enables derivation of fast learning rates.

Building upon this framework, we prove that if the loss has finite moments up to order $r$ large enough and if the hypothesis class satisfies the regularity conditions described above, then learning rate faster than $O(n^{-1/2})$ can be obtained. Moreover, depending on $r$ and the multi-scale Bernstein's powers, the learning rate can be arbitrarily close to the optimal rate $O(n^{-1})$. We then verify these assumptions and derive fast learning rates for the $k$-mean clustering algorithm and prove that if the distribution of observations has finite moments up to order $r$ and satisfies the Pollard's regularity conditions, then fast learning rate can be derived. The result can be viewed as an extension of the result from Antos et al. [2005] and Levrard [2013] to cases when the source distribution has unbounded support, and produces a more favorable convergence rate than that of Telgarsky and Dasgupta [2013] under similar settings.

## 2 Mathematical framework

Let the hypothesis class $\mathcal{F}$ be a class of functions defined on some measurable space $\mathcal{X}$ with values in $\mathbb{R}$. Let $Z = (X, Y)$ be a random variable taking values in $\mathcal{Z} = \mathcal{X} \times \mathcal{Y}$ with probability distribution $P$ where $\mathcal{Y} \subset \mathbb{R}$. The loss $\ell : \mathcal{Z} \times \mathcal{F} \to \mathbb{R}^+$ is a non-negative function. For a hypothesis $f \in \mathcal{F}$ and $n$ iid samples $\{Z_1, Z_2, \ldots, Z_n\}$ of $Z$, we define

$$P\ell(f) = \mathbb{E}_{Z \sim P}[\ell(Z, f)] \qquad \text{and} \qquad P_n\ell(f) = \frac{1}{n} \sum_{i=1}^{n} \ell(Z_i, f).$$

For unsupervised learning frameworks, there is no output ($\mathcal{Y} = \emptyset$) and the loss has the form $\ell(X, f)$ depending on applications. Nevertheless, $P\ell(f)$ and $P_n\ell(f)$ can be defined in a similar manner. We will abuse the notation to denote the losses $\ell(Z, f)$ by $\ell(f)$. We also denote the optimal hypothesis $f^*$ be any function for which $P\ell(f^*) = \inf_{f \in \mathcal{F}} P\ell(f) := P^*$ and consider the empirical risk minimizer (ERM) estimator $\hat{f}_n = \arg\min_{f \in \mathcal{F}} P_n\ell(f)$.

We recall that heavy-tailed distributions are probability distributions whose tails are not exponentially bounded. Rigorously, the distribution of a random variable $V$ is said to have a heavy right tail if $\lim_{v \to \infty} e^{\lambda v} \mathbb{P}[V > v] = \infty$ for all $\lambda > 0$ and the definition is similar for heavy left tail. A learning problem is said to be with heavy-tailed loss if the distribution of $\ell(f)$ has heavy tails from some or all hypotheses $f \in \mathcal{F}$.

For a pseudo-metric space $(G, d)$ and $\epsilon > 0$, we denote by $\mathcal{N}(\epsilon, G, d)$ the *covering number* of $(G, d)$; that is, $\mathcal{N}(\epsilon, G, d)$ is the minimal number of balls of radius $\epsilon$ needed to cover G. The *universal metric entropy* of $G$ is defined by $\mathcal{H}(\epsilon, G) = \sup_Q \log \mathcal{N}(\epsilon, G, L^2(Q))$, where the supremum is taken over the set of all probability measures $Q$ concentrated on some finite subset of $G$. For convenience, we define $\mathcal{G} = \ell \circ \mathcal{F}$ the class of all functions $g$ such that $g = \ell(f)$ for some $f \in \mathcal{F}$ and denote by $\mathcal{F}_\epsilon$ a finite subset of $\mathcal{F}$ such that $\mathcal{G}$ is contained in the union of balls of radius $\epsilon$ with centers in $\mathcal{G}_\epsilon = \ell \circ \mathcal{F}_\epsilon$. We refer to $\mathcal{F}_\epsilon$ and $\mathcal{G}_\epsilon$ as an $\epsilon$-net of $\mathcal{F}$ and $\mathcal{G}$, respectively.

To enable the analyses of fast rates for learning problems with heavy-tailed losses, throughout the paper, we impose the following regularity conditions on $\mathcal{F}$ and $\ell$.

**Assumption 2.1** (Multi-scale Bernstein's condition). *Define $\mathcal{F}^* = \arg\min_{\mathcal{F}} P\ell(f)$. There exist a finite partition of $\mathcal{F} = \cup_{i \in I} \mathcal{F}_i$, positive constants $B = \{B_i\}_{i \in I}$, constants $\gamma = \{\gamma_i\}_{i \in I}$ in $(0, 1]$, and $f^* = \{f_i^*\}_{i \in I} \subset \mathcal{F}^*$ such that $\mathbb{E}[(\ell(f) - \ell(f_i^*))^2] \leq B_i \left(\mathbb{E}[\ell(f) - \ell(f_i^*)]\right)^{\gamma_i}$ for all $i \in I$ and $f \in \mathcal{F}_i$.*

**Assumption 2.2** (Entropy bounds). *The hypothesis class $\mathcal{F}$ is separable and there exist $\mathcal{C} \geq 1$, $K \geq 1$ such that $\forall \epsilon \in (0, K]$, the $L_2(P)$-covering numbers and the universal metric entropies of $\mathcal{G}$ are bounded as $\log \mathcal{N}(\epsilon, \mathcal{G}, L_2(P)) \leq \mathcal{C} \log(K/\epsilon)$ and $\mathcal{H}(\epsilon, \mathcal{G}) \leq \mathcal{C} \log(K/\epsilon)$.*

**Assumption 2.3** (Integrability of the envelope function). *There exists $W > 0, r \geq \mathcal{C} + 1$ such that $\left(\mathbb{E} \sup_{g \in \mathcal{G}} |g|^r\right)^{1/r} \leq W$.*

The multi-scale Bernstein's condition is more general than the Bernstein's condition. This entails that the multi-scale Bernstein's condition holds whenever the Bernstein's condition does, thus al-

lows us to consider a larger class of problems. In other words, our results are also valid with the Bernstein's condition. The multi-scale Bernstein's condition is more proper to study unbounded losses since it is able to separately consider the behaviors of the risk function on microscopic and macroscopic scales, for which the distinction can only be observed in an unbounded setting.

We also recall that if $\mathcal{G}$ has finite VC-dimension, then Assumption 2.2 is satisfied [Boucheron et al., 2013, Bousquet et al., 2004]. Both Bernstein's condition and the assumption of separable parametric hypothesis class are standard assumptions frequently used to obtain faster learning rates in agnostic settings. A review about the Bernstein's condition and its applications is Mendelson [2008], while fast learning rates for bounded losses on hypothesis classes satisfying Assumptions 2.2 were previously studied in Mehta and Williamson [2014] under the stochastic mixability condition. Fast learning rate for hypothesis classes with envelope functions were studied in Lecué and Mendelson [2012], but under a much stronger assumption that the envelope function is sub-exponential.

Under these assumptions, we illustrate that fast rates for heavy-tailed losses can be obtained. Throughout the analyses, two recurrent analytical techniques are worth mentioning. The first comes from the simple observation that in the standard derivation of fast learning rates for bounded losses, the boundedness assumption is used in multiple places only to provide reverse-Holder-type inequalities, where the $L_2$-norm are upper bounded by the $L_1$-norm. This use of the boundedness assumption can be simply relieved by the assumption that the $L_r$-norm of the loss is bounded, which implies

$$\|u\|_{L_2} \leq \|u\|_{L_1}^{(r-2)/(2r-2)} \|u\|_{L_r}^{r/(2r-2)}.$$

The second technique relies on the following results of Lederer and van de Geer [2014] on concentration inequalities for suprema of empirical unbounded processes.

**Lemma 2.1.** *If* $\{V_k : k \in \mathcal{K}\}$ *is a countable family of non-negative functions such that*

$$\mathbb{E} \sup_{k \in \mathcal{K}} |V_k|^r \leq M^r \qquad \sigma^2 = \sup_{k \in \mathcal{K}} \mathbb{E} V_k^2 \qquad \text{and} \qquad V := \sup_{k \in \mathcal{K}} P_n V_k,$$

*then for all* $\zeta, x > 0$, *we have*

$$\mathbb{P}[V \geq (1+\zeta)\mathbb{E}V + x] \leq \min_{1 \leq l \leq r} (1/x)^l \left[ \left( 64/\zeta + \zeta + 7 \right) (l/n)^{1-l/r} M + 4\sigma\sqrt{l/n} \right)^l \right].$$

An important notice from this result is that the failure probability is a polynomial in the deviation $x$. As we will see later, for a given level of confidence $\delta$, this makes the constant in the convergence rate a polynomial function of $(1/\delta)$ instead of $\log(1/\delta)$ as in sub-exponential cases. Thus, more careful examinations of the order of the failure probability are required for the derivation of any generalization bound with heavy-tailed losses.

## 3  Fast learning rates with heavy-tailed losses

The derivation of fast learning rate with heavy tailed losses proceeds as follows. First, we will use the assumption of integrable envelope function to prove a localization-based result that allows us to reduce the analyses from the separable parametric classes $\mathcal{F}$ to its finite $\epsilon$-net $\mathcal{F}_\epsilon$. The multi-scale Bernstein's condition is then employed to derive a fast-rate inequality that helps distinguish the optimal hypothesis from alternative hypotheses in $\mathcal{F}_\epsilon$. The two results are then combined to obtain fast learning rates.

### 3.1  Preliminaries

Throughout this section, let $\mathcal{G}_\epsilon$ be an $\epsilon$-net for $\mathcal{G}$ in the $L_2(P)$-norm, with $\epsilon = n^{-\beta}$ for some $1 \geq \beta > 0$. Denote by $\pi : \mathcal{G} \to \mathcal{G}_\epsilon$ an $L_2(P)$-metric projection from $\mathcal{G}$ to $\mathcal{G}_\epsilon$. For any $g_0 \in \mathcal{G}_\epsilon$, we denote $\mathcal{K}(g_0) = \{|g_0 - g| : g \in \pi^{-1}(g_0)\}$. We have

(i) the constant zero function is an element of $\mathcal{K}(g_0)$,

(ii) $\mathbb{E}[\sup_{u \in \mathcal{K}(g_0)} |u|^r] \leq (2W)^r$; and $\sup_{u \in \mathcal{K}(g_0)} \|u\|_{L_2(P)} \leq \epsilon$,

(iii) $\mathcal{N}(t, \mathcal{K}(g_0), L_2(P)) \leq (K/t)^C$ for all $t > 0$.

Given a sample $Z = (Z_1, \ldots, Z_n)$, we denote by $\mathcal{K}_Z$ the projection of $\mathcal{K}(g_0)$ onto the sample $Z$ and by $D(\mathcal{K}_Z)$ half of the radius of $(\mathcal{K}_Z, \|\cdot\|_2)$, that is $D(\mathcal{K}_Z) = \sup_{u,v \in \mathcal{K}_Z} \|u - v\|/4$. We have the following preliminary lemma, for which the proofs are provided in the Appendix.

**Lemma 3.1.** $\frac{2}{\sqrt{n}} \mathbb{E} D(\mathcal{K}_Z) \leq \left( \epsilon + \mathbb{E} \sup_{u \in \mathcal{K}(g_0)} (P_n - P)u \right)^{\frac{r-2}{2(r-1)}} (2W)^{\frac{r}{2(r-1)}}.$

**Lemma 3.2.** *Given $0 < \nu < 1$, there exist constant $C_1, C_2 > 0$ depending only on $\nu$ such that for all $x > 0$, if $x \leq ax^\nu + b$ then $x \leq C_1 a^{1/(1-\nu)} + C_2 b$.*

**Lemma 3.3.** *Define*

$$A(l, r, \beta, \mathcal{C}, \alpha) = \max \left\{ l^2/r - (1 - \beta)l + \beta\mathcal{C}, [\beta(1 - \alpha/2) - 1/2]l + \beta\mathcal{C} \right\}. \tag{3.1}$$

*Assuming that $r \geq 4\mathcal{C}$ and $\alpha \leq 1$, if we choose $l = r(1 - \beta)/2$ and*

$$0 < \beta < (1 - 2\sqrt{\mathcal{C}/r})/(2 - \alpha), \tag{3.2}$$

*then $1 \leq l \leq r$ and $A(l, r, \beta, \mathcal{C}, \alpha) < 0$. This also holds if $\alpha \geq 1$ and $0 < \beta < 1 - 2\sqrt{\mathcal{C}/r}$.*

## 3.2 Local analysis of the empirical loss

The preliminary lemmas enable us to locally bound $\mathbb{E} \sup_{u \in \mathcal{K}(g_0)} (P_n - P)u$ as follows:

**Lemma 3.4.** *If $\beta < (r - 1)/r$, there exists $c_1 > 0$ such that $\mathbb{E} \sup_{u \in \mathcal{K}(g_0)} (P_n - P)u \leq c_1 n^{-\beta}$ for all $n$.*

*Proof.* Without loss of generality, we assume that $\mathcal{K}(g_0)$ is countable. The arguments to extend the bound from countable classes to separable classes are standard (see, for example, Lemma 12 of Mehta and Williamson [2014]). Denote $\bar{Z} = \sup_{u \in \mathcal{K}(g_0)} (P_n - P)u$ and let $\epsilon = 1/n^\beta$, $\mathbf{R} = (R_1, R_2, \ldots, R_n)$ be iid Rademacher random variables, using standard results about symmetrization and chaining of Rademacher process (see, for example, Corollary 13.2 in Boucheron et al. [2013]), we have

$$n\mathbb{E} \sup_{u \in \mathcal{K}(g_0)} (P_n - P)g \leq 2\mathbb{E} \left( \mathbb{E}_{\mathbf{R}} \sup_{u \in \mathcal{K}(g_0)} \sum_{j=1}^n R_j u(X_j) \right)$$

$$\leq 24\mathbb{E} \int_0^{D(\mathcal{K}_X) \vee \epsilon} \sqrt{\log \mathcal{N}(t, \mathcal{K}_X, \|\cdot\|_2)} dt \leq 24\mathbb{E} \int_0^{D(\mathcal{K}_X) \vee \epsilon} \sqrt{\mathcal{H}(t/\sqrt{n}, \mathcal{K}(g_0))} dt,$$

where $\mathbb{E}_{\mathbf{R}}$ denotes the expected value with respect to the random variables $R_1, R_2, \ldots, R_n$. By Assumption 2.2, we deduce that

$$n\mathbb{E}\bar{Z} \leq C_0(K, n, \sigma, \mathcal{C})(\epsilon + \mathbb{E} D(\mathcal{K}_X)) \qquad \text{where} \qquad C_0 = \mathcal{O}(\sqrt{\log n}).$$

If we define

$$x = \epsilon + \mathbb{E}\bar{Z}, \ b = C_0\epsilon/n = \mathcal{O}(\sqrt{\log n}/n^{\beta+1}), \ a = C_0 n^{-1/2}(2W)^{\frac{r}{2(r-1)}}/2 = \mathcal{O}(\sqrt{\log n}/\sqrt{n}),$$

then by Lemma 3.1, we have $x \leq ax^{(r-2)/(2r-2)} + b + \epsilon$. Using lemma 3.2, we have

$$x \leq C_1 a^{2(r-1)/r} + C_2(b + \epsilon) \leq C_3 n^{-\beta},$$

which completes the proof. $\qquad \square$

**Lemma 3.5.** *Assuming that $r \geq 4\mathcal{C}$, if $\beta < 1 - 2\sqrt{\mathcal{C}/r}$, there exist $c_1, c_2 > 0$ such that for all $n$ and $\delta > 0$*

$$\sup_{u \in \mathcal{K}(g_0)} P_n u \leq \left( 9c_1 + (c_2/\delta)^{1/[r(1-\beta)]} \right) n^{-\beta} \qquad \forall g_0 \in \mathcal{G}_\epsilon$$

*with probability at least $1 - \delta$.*

*Proof.* Denote $Z = \sup_{u\in\mathcal{K}(g_0)} P_n u$ and $\bar{Z} = \sup_{u\in\mathcal{K}(g_0)} (P_n - P)u$. We have

$$Z = \sup_{u\in\mathcal{K}(g_0)} P_n u \leq \bar{Z} + \sup_{u\in\mathcal{K}(g_0)} Pu \leq \bar{Z} + \sup_{u\in\mathcal{K}(g_0)} \|u\|_{L^2(P)} = \bar{Z} + \epsilon.$$

Applying Lemma 2.1 for $\zeta = 8$ and $x = y/n^\beta$ for $\bar{Z}$, using the facts that

$$\sigma = \sup_{u\in\mathcal{K}_{g_0}} \sqrt{\mathbb{E}[u(X)]^2} \leq \epsilon = 1/n^\beta, \qquad \text{and} \qquad \mathbb{E}[\sup_{u\in\mathcal{K}_{g_0}} |u|^r] \leq (2W)^r,$$

we have

$$\mathbb{P}\left[\bar{Z} \geq 9\mathbb{E}\bar{Z} + y/n^\beta\right] \leq \min_{1\leq l\leq r} y^{-l}\left[\left(46\,(l/n)^{1-l/r}\, n^\beta W + 4\sqrt{l/n}\right)^l\right] := \phi(y,n).$$

To provide a union bound for all $g_0 \in \mathcal{G}_\epsilon$, we want the total failure probability $\phi(y,n)(n^\beta K)^{\mathcal{C}} \leq \delta$. This failure probability, as a function of $n$, is of order $A(l,r,\beta,\mathcal{C},\alpha)$ (as define in Lemma 3.3) with $\alpha = 2$. By choosing $l = r(1-\beta)/2$ and $\beta < 1 - 2\sqrt{\mathcal{C}/r}$, we deduce that there exist $c_2, c_3 > 0$ such that $\phi(y,n)(n^\beta K)^{\mathcal{C}} \leq c_2/(n^{c_3}y^l) \leq c_2/y^{r(1-\beta)/2}$. The proof is completed by choosing $y = (c_2/\delta)^{2/[r(1-\beta)]}$ and using the fact that $\mathbb{E}\bar{Z} \leq c_1/n^\beta$ (note that $1 - 2\sqrt{\mathcal{C}/r} \leq (r-1)/r$ and we can apply Lemma 3.4 to get the bound). $\square$

A direct consequence of this Lemma is the following localization-based result.

**Theorem 3.1** (Local analysis). *Under Assumptions 2.1, 2.2 and 2.3, let $\mathcal{G}_\epsilon$ be a minimal $\epsilon$-net for $\mathcal{G}$ in the $L_2(P)$-norm, with $\epsilon = n^{-\beta}$ where $\beta < 1 - 2\sqrt{\mathcal{C}/r}$. Then there exist $c_1, c_2 > 0$ such that for all $\delta > 0$,*

$$P_n g \geq P_n(\pi(g)) - \left(9c_1 + (c_2/\delta)^{2/[r(1-\beta)]}\right)n^{-\beta} \qquad \forall g \in \mathcal{G}$$

*with probability at least $1 - \delta$.*

### 3.3 Fast learning rates with heavy-tailed losses

**Theorem 3.2.** *Given $a_0, \delta > 0$. Under the multi-scale $(B, \gamma, I)$-Bernstein's condition and the assumption that $r \geq 4\mathcal{C}$, consider*

$$0 < \beta < (1 - 2\sqrt{\mathcal{C}/r})/(2 - \gamma_i) \qquad \forall i \in I. \tag{3.3}$$

*Then there exist $N_{a_0,\delta,r,B,\gamma} > 0$ such that $\forall f \in \mathcal{F}_\epsilon$ and $n \geq N_{a_0,\delta,r,B,\gamma}$, we have*

$$P\ell(f) - P^* \geq a_0/n^\beta \qquad \text{implies} \qquad \exists f^* \in \mathcal{F}^*: \quad P_n\ell(f) - P_n\ell(f^*) \geq a_0/(4n^\beta)$$

*with probability at least $1 - \delta$.*

*Proof.* Define $a = [P\ell(f) - P^*]n^\beta$. Assuming that $f \in \mathcal{F}_i$, applying Lemma 2.1 for $\zeta = 1/2$ and $x = a/4n^\beta$ for a single hypothesis $f$, we have

$$\mathbb{P}\left[P_n\ell(f) - P_n\ell(f_i^*) \leq (P\ell(f) - P\ell(f_i^*))/4\right] \leq h(a,n)$$

where

$$h(a,n,i) = \min_{1\leq l\leq r} (4/a)^l \left(50n^\beta\,(l/n)^{1-l/r}\,W + 4n^\beta B_i a^{\gamma_i/2}/n^{\beta\gamma_i/2}\sqrt{l/n}\right)^l$$

using the fact that $\sigma^2 = \mathbb{E}[\ell(f) - \ell(f_i^*)]^2 \leq B_i\left[\mathbb{E}(\ell(f) - \ell(f_i^*))\right]^{\gamma_i} = B_i a^{\gamma_i}/n^{\beta\gamma_i}$ if $f \in \mathcal{F}_i$. Since $\gamma_i \leq 1$, $h(a,n,i)$ is a non-increasing function in $a$. Thus,

$$\mathbb{P}\left[P_n\ell(f) - P_n\ell(f_i^*) \leq (P\ell(f) - P\ell(f_i^*))/4\right] \leq h(a_0,n,i).$$

To provide a union bound for all $f \in \mathcal{F}_\epsilon$ such that $P\ell(f) - P\ell(f_i^*) \geq a_0/n^\beta$, we want the total failure probability to be small. This is guaranteed if $h(a_0,n,i)(n^\beta K)^{\mathcal{C}} \leq \delta$. This failure probability, as a function of $n$, is of order $A(l,r,\beta,\mathcal{C},\gamma_i)$ as defined in equation (3.1). By choosing $r, l$ as in Lemma 3.3 and $\beta$ as in equation (3.3), we have $1 \leq l \leq r$ and $A(l,r,\beta,\mathcal{C},\gamma_i) < 0$ for all $i$. Thus, there exists $c_4, c_5, c_6 > 0$ such that

$$h(a_0,n,i)(n^\beta K)^{\mathcal{C}} \leq c_6 a_0^{-c_5(1-\gamma_i/2)}n^{-c_4} \qquad \forall n, i.$$

Hence, when $n \geq N_{a,\delta,r,B,\gamma} = \left(c_6\delta a_0^{-c_5(1-\tilde\gamma/2)}\right)^{1/c_4}$ where $\tilde\gamma = \max\{\gamma\}1_{\{a_0\geq 1\}} + \min\{\gamma\}1_{\{a_0<1\}}$, we have: $\forall f \in \mathcal{F}_\epsilon, P\ell(f) - P^* \geq a_0/n^\beta$ implies $\exists f^* \in \mathcal{F}^*, P_n\ell(f) - P_n\ell(f^*) \geq a_0/(4n^\beta)$ with probability at least $1 - \delta$. $\square$

**Theorem 3.3.** *Under Assumptions* 2.1, 2.2 *and* 2.3, *consider* $\beta$ *as in equation* (3.3) *and* $c_1, c_2$ *as in previous theorems. For all* $\delta > 0$, *there exists* $N_{\delta,r,B,\gamma}$ *such that if* $n \geq N_{\delta,r,B,\gamma}$, *then*

$$P\ell(\hat{f}_z) \leq P\ell(f^*) + \left(36c_1 + 1 + 4\,(2c_2/\delta)^{2/[r(1-\beta)]}\right) n^{-\beta}$$

*with probability at least* $1 - \delta$.

*Proof of Theorem* 3.3. Let $\mathcal{F}_\epsilon$ by an $\epsilon$-net of $\mathcal{F}$ with $\epsilon = 1/n^\beta$ such that $f^* \in \mathcal{F}_\epsilon$. We denote the projection of $\hat{f}_z$ to $\mathcal{F}_\epsilon$ by $f_1 = \pi(\hat{f}_z)$. For a given $\delta > 0$, define

$$A_1 = \left\{ \exists f \in \mathcal{F} : P_n f \leq P_n(\pi(f)) - \left(9c_1 + (c_3/\delta)^{2/[r(1-\beta)]}\right) n^{-\beta} \right\},$$

$$A_2 = \left\{ \exists f \in \mathcal{F}_\epsilon : P_n\ell(\pi(f)) - P_n\ell(f^*) \leq a_0/(4n^\beta) \text{ and } P\ell(\pi(f)) - P\ell(f^*) \geq a_0/n^\beta \right\},$$

where $c_1, c_2$ is defined as in previous theorem, $a_0/4 = 9c_1 + (c_3/\delta)^{2/[r(1-\beta)]}$ and $n \geq N_{a_0,\delta,r,\gamma}$. We deduce that $A_1$ and $A_2$ happen with probability at most $\delta$. On the other hand, under the event that $A_1$ and $A_2$ do not happen, we have

$$P_n\ell(f_1) \leq P_n\ell(\hat{f}_z) + \left(9c_1 + (c_3/\delta)^{2/[r(1-\beta)]}\right) n^{-\beta} \leq P_n\ell(f^*) + a_0/(4n^\beta).$$

By definition of $\mathcal{F}_\epsilon$, we have $P\ell(\hat{f}_z) \leq P\ell(f_1) + \epsilon \leq P\ell(f^*) + (a_0 + 1)/n^\beta$. $\square$

## 3.4 Verifying the multi-scale Bernstein's condition

In practice, the most difficult condition to verify for fast learning rates is the multi-scale Bernstein's condition. We derive in this section some approaches to verify the condition. We first extend the result of Mendelson [2008] to prove that the (standard) Bernstein's condition is automatically satisfied for functions that are relatively far way from $f^*$ under the integrability condition of the envelope function (proof in the Appendix). We recall that $R(f) = \mathbb{E}\ell(f)$ is referred to as the risk function.

**Lemma 3.6.** *Under Assumption* 2.3, *we define* $M = W^{r/(r-2)}$ *and* $\gamma = (r-2)/(r-1)$. *Then, if* $\alpha > M$ *and* $R(f) \geq \alpha/(\alpha - M)R(f^*)$, *then* $\mathbb{E}(\ell(f) - \ell(f^*))^2 \leq 2\alpha^\gamma \mathbb{E}(\ell(f) - \ell(f^*))^\gamma$.

This allows us to derive the following result, for which the proof is provided in the Appendix.

**Lemma 3.7.** *If* $\mathcal{F}$ *is a subset of a vector space with metric* $d$ *and the risk function* $R(f) = \mathbb{E}\ell(f)$ *has a unique minimizer on* $\mathcal{F}$ *at* $f^*$ *in the interior of* $\mathcal{F}$ *and*

   *(i) There exists* $L > 0$ *such that* $\mathbb{E}(\ell(f) - \ell(g))^2 \leq Ld(f,g)^2$ *for all* $f, g \in \mathcal{F}$.

   *(ii) There exists* $m \geq 2$, $c > 0$ *and a neighborhood* $U$ *around* $f^*$ *such that*

      $R(f) - R(f^*) \geq cd(f, f^*)^m$ *for all* $f \in U$.

*Then the multi-scale Bernstein's condition holds for* $\gamma = ((r-2)/(r-1), 2/m)$.

**Corollary 3.1.** *Suppose that* $(\mathcal{F}, d)$ *is a pseudo-metric space,* $\ell$ *satisfies condition (i) in Lemma* 3.7 *and the risk function is strongly convex with respect to* $d$, *then the Bernstein's condition holds with* $\gamma = 1$.

**Remark 3.1.** *If the risk function is analytic at* $f^*$, *then condition (ii) in Lemma* 3.7 *holds. Similarly, if the risk function is continuously differentiable up to order 2 and the Hessian of* $R(f)$ *is positive definite at* $f^*$, *then condition (ii) is valid with* $m = 2$.

**Corollary 3.2.** *If the risk function* $R(f) = \mathbb{E}\ell(f)$ *has a finite number of global minimizers* $f_1, f_2, \ldots, f_k$, $\ell$ *satisfies condition (i) in Lemma* 3.7 *and there exists* $m_i \geq 2$, $c_i > 0$ *and neighborhoods* $U_i$ *around* $f_i$ *such that* $R(f) - R(f_i) \geq c_i d(f, f_i)^{m_i}$ *for all* $f \in U_i, i = 1, \ldots, k$, *then the multi-scale Bernstein's condition holds for* $\gamma = ((r-2)/(r-1), 2/m_1, \ldots, 2/m_k)$.

## 3.5 Comparison to related work

Theorem 3.3 dictates that under our settings, the problem of learning with heavy-tailed losses can obtain convergence rates up to order

$$\mathcal{O}\left(n^{-(1-2\sqrt{C/r})/(2-\min\{\gamma\})}\right) \tag{3.4}$$

where $\gamma$ is the multi-scale Bernstein's order and $r$ is the degree of integrability of the loss. We recall that convergence rate of $\mathcal{O}(n^{-1/(2-\gamma)})$ is obtained in Mehta and Williamson [2014] under the same setting but for bounded losses. (The analysis there was done under the $\gamma$-weakly stochastic mixability condition, which is equivalent with the standard $\gamma$-Bernstein's condition for bounded losses [van Erven et al., 2015]). We note that if the loss is bounded, $r = \infty$ and (3.4) reduces to the convergence rate obtained in Mehta and Williamson [2014].

Fast learning rates for unbounded loses are previously derived in Lecué and Mendelson [2013] for sub-Gaussian losses and in Lecué and Mendelson [2012] for hypothesis classes that have sub-exponential envelope functions. In Lecué and Mendelson [2013], the Bernstein's condition is not directly imposed, but is replaced by condition (ii) of Lemma 3.7 with $m = 2$ on the whole hypothesis class, while the assumption of sub-Gaussian hypothesis class validates condition (i). This implies the standard Bernstein's condition with $\gamma = 1$ and makes the convergence rate $\mathcal{O}(n^{-1})$ consistent with our result (note that for sub-Gaussian losses, $r$ can be chosen arbitrary large). The analysis of Lecué and Mendelson [2012] concerns about *non-exact oracle inequalities* (rather than the *sharp oracle inequalities* we investigate in this paper) and can not be directly compared with our results.

# 4    Application: $k$-means clustering with heavy-tailed source distributions

$k$-means clustering is a method of vector quantization aiming to partition $n$ observations into $k \geq 2$ clusters in which each observation belongs to the cluster with the nearest mean. Formally, let $X$ be a random vector taking values in $\mathbb{R}^d$ with distribution $P$. Given a codebook (set of $k$ cluster centers) $C = \{y_i\} \in (\mathbb{R}^d)^k$, the distortion (loss) on an instant $x$ is defined as $\ell(C, x) = \min_{y_i \in C} \|x - y_i\|^2$ and $k$-means clustering method aims at finding a minimizer $C^*$ of $R(\ell(C)) = P\ell(C)$ via minimizing the empirical distortion $P_n \ell(C)$.

The rate of convergence of $k$-means clustering has drawn considerable attention in the statistics and machine learning literatures [Pollard, 1982, Bartlett et al., 1998, Linder et al., 1994, Ben-David, 2007]. Fast learning rates for $k$-means clustering ($\mathcal{O}(1/n)$) have also been derived by Antos et al. [2005] in the case when the source distribution is supported on a finite set of points, and by Levrard [2013] under the assumptions that the source distribution has bounded support and satisfies the so-called Pollard's regularity condition, which dictates that $P$ has a continuous density with respect to the Lebesgue measure and the Hessian matrix of the mapping $C \to R(C)$ is positive definite at $C^*$. Little is known about the finite-sample performance of empirically designed quantizers under possibly heavy-tailed distributions. In Telgarsky and Dasgupta [2013], a convergence rate of $\mathcal{O}(n^{-1/2+2/r})$ are derived, where $r$ is the number of moments of $X$ that are assumed to be finite. Brownlees et al. [2015] uses some robust mean estimators to replace empirical means and derives a convergence rate of $\mathcal{O}(n^{-1/2})$ assuming only that the variance of $X$ is finite.

The results from previous sections enable us to prove that with proper setting, the convergence rate of $k$-means clustering for heavy-tailed source distributions can be arbitrarily close to $\mathcal{O}(1/n)$. Following the framework of Brownlees et al. [2015], we consider

$$\mathcal{G} = \{\ell(C, x) = \min_{y_i \in C} \|x - y_i\|^2, C \in \mathcal{F} = (-\rho, \rho)^{d \times k}\}$$

for some $\rho > 0$ with the regular Euclidean metric. We let $C^*, \hat{C}_n$ be defined as in the previous sections.

**Theorem 4.1.** *If $X$ has finite moments up to order $r \geq 4k(d+1)$, $P$ has a continuous density with respect to the Lebesgue measure, the risk function has a finite number of global minimizers and the Hessian matrix of $C \to R(C)$ is positive definite at the every optimal $C^*$ in the interior of $\mathcal{F}$, then for all $\beta$ that satisfies*

$$0 < \beta < \frac{r-1}{r}(1 - 2\sqrt{k(d+1)/r}),$$

*there exists $c_1, c_2 > 0$ such that for all $\delta > 0$, with probability at least $1 - \delta$, we have*

$$R(\hat{C}_n) - R(C^*) \leq \left(c_1 + 4\,(c_2/\delta)^{2/r}\right) n^{-\beta}$$

*Moreover, when $r \to \infty$, $\beta$ can be chosen arbitrarily close to 1.*

*Proof.* We have

$$\left(\mathbb{E}\sup_{C\in\mathcal{F}}\ell(C,X)^r\right)^{1/r} \leq \left(\frac{1}{2^r}\mathbb{E}[\|X\|^2+\rho^2]^r\right)^{1/r} \leq \left(\frac{1}{2}\mathbb{E}\|X\|^{2r}+\frac{1}{2}\rho^{2r}\right)^{1/r} \leq W < \infty,$$

while standard results about VC-dimension of k-means clustering hypothesis class guarantees that $\mathcal{C} \leq k(d+1)$ [Linder et al., 1994]. On the other hand, we can verify that

$$\mathbb{E}[\ell(C,X)-\ell(C',X)]^2 \leq L_\rho\|C-C'\|_2^2,$$

which validates condition (i) in Lemma 3.7. The fact that the Hessian matrix of $C \to R(C)$ is positive definite at $C^*$ prompts $R(\hat{C}_n)-R(C^*) \geq c\|\hat{C}_n-C^*\|^2$ for some $c > 0$ in a neighborhood $U$ around any optimal codebook $C^*$. Thus, Lemma 3.6 confirms the multi-scale Bernstein's condition with $\gamma = ((r-2)/(r-1),1,\ldots,1)$. The inequality is then obtained from Theorem 3.3. $\quad\square$

## 5  Discussion and future work

We have shown that fast learning rates for heavy-tailed losses can be obtained for hypothesis classes with an integrable envelope when the loss satisfies the multi-scale Bernstein's condition. We then verify those conditions and obtain new convergence rates for $k$-means clustering with heavy-tailed losses. The analyses extend and complement existing results in the literature from both theoretical and practical points of view. We also introduce a new fast-rate assumption, the multi-scale Bernstein's condition, and provide a clear path to verify the assumption in practice. We believe that the multi-scale Bernstein's condition is the proper assumption to study fast rates for unbounded losses, for its ability to separate the behaviors of the risk function on microscopic and macroscopic scales, for which the distinction can only be observed in an unbounded setting.

There are several avenues for improvement. First, we would like to consider hypothesis class with polynomial entropy bounds. Similarly, the condition of independent and identically distributed observations can be replaced with mixing properties [Steinwart and Christmann, 2009, Hang and Steinwart, 2014, Dinh et al., 2015]. While the condition of integrable envelope is an improvement from the condition of sub-exponential envelope previously investigated in the literature, it would be interesting to see if the rates retain under weaker conditions, for example, the assumption that the $L^r$-diameter of the hypothesis class is bounded [Cortes et al., 2013]. Finally, the recent work of Brownlees et al. [2015], Hsu and Sabato [2016] about robust estimators as alternatives of ERM to study heavy-tailed losses has yielded more favorable learning rates under weaker conditions, and we would like to extend the result in this paper to study such estimators.

## Acknowledgement

Vu Dinh was supported by DMS-1223057 and CISE-1564137 from the National Science Foundation and U54GM111274 from the National Institutes of Health. Lam Si Tung Ho was supported by NSF grant IIS 1251151.

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
