[Supplementary Material]

# Fast learning rates with heavy-tailed losses

**Vu Dinh**[1]    **Lam Si Tung Ho**[2]    **Duy Nguyen**[3]    **Binh T. Nguyen**[4]
[1]Program in Computational Biology, Fred Hutchinson Cancer Research Center
[2]Department of Biostatistics, University of California, Los Angeles
[3]Department of Statistics, University of Wisconsin-Madison
[4]Department of Computer Science, University of Science, Vietnam

## Appendix

*Proof of Lemma 3.1.* Define $q = r-1$; $p = \frac{r-1}{r-2}$. Note that $u$ is non-negative, by Holder's inequality we have

$$P_n u^2 \leq (P_n u)^{1/p} (P_n u^r)^{1/q}$$

which implies

$$\sup_{u \in \mathcal{K}(g_0)} \sqrt{P_n u^2} \leq \left( \sup_{u \in \mathcal{K}(g_0)} P_n u \right)^{\frac{1}{2p}} \left( \sup_{u \in \mathcal{K}(g_0)} P_n u^r \right)^{\frac{1}{2q}}.$$

If we rewrite this inequality as $y \leq h^{1/p} k^{1/q}$, then

$$\mathbb{E}y \leq \mathbb{E}[h^{1/p} k^{1/q}] \leq (\mathbb{E}h)^{1/p} (\mathbb{E}k)^{1/q}. \tag{0.1}$$

Since $Z_i$'s are independently identically distributed, we have

$$\mathbb{E} \sup_{u \in \mathcal{K}(g_0)} P_n u^r = \mathbb{E} \sup_{u \in \mathcal{K}(g_0)} \frac{u^r(Z_1) + \ldots + u^r(Z_n)}{n} \leq \mathbb{E} \frac{1}{n} \sum_{i=1}^n \sup_{u \in \mathcal{K}(g_0)} u^r(Z_i) = \mathbb{E} \sup_{u \in \mathcal{K}(g_0)} u^r(Z_1).$$

This implies

$$\mathbb{E} \sup_{u \in \mathcal{K}(g_0)} \sqrt{P_n u^r} \leq \sqrt{\mathbb{E} \sup_{u \in \mathcal{K}(g_0)} P_n u^r} \leq \left( \mathbb{E} \sup_{u \in \mathcal{K}(g_0)} u^r(Z_1) \right)^{1/2}. \tag{0.2}$$

On the other hand, using the notation $\bar{P}_n = P_n - P$, we have

$$\mathbb{E} \sup_{u \in \mathcal{K}(g_0)} P_n u \leq \mathbb{E} \sup_{u \in \mathcal{K}(g_0)} \bar{P}_n u + \sup_{u \in \mathcal{K}(g_0)} Pu \leq \mathbb{E} \sup_{u \in \mathcal{K}(g_0)} \bar{P}_n u + \sup_{u \in \mathcal{K}(g_0)} \sqrt{Pu^2} \leq \mathbb{E} \sup_{u \in \mathcal{K}(g_0)} \bar{P}_n u + \epsilon. \tag{0.3}$$

Combining (0.1), (0.2) and (0.3), we deduce that

$$\frac{2}{\sqrt{n}} \mathbb{E}D(\mathcal{K}_X) \leq \mathbb{E} \sup_{u \in \mathcal{K}(g_0)} \sqrt{P_n u^2} \leq \left( \epsilon + \mathbb{E} \sup_{u \in \mathcal{K}(g_0)} (P_n - P)u \right)^{\frac{r-2}{2(r-1)}} (2W)^{\frac{r}{2(r-1)}}.$$

$\square$

*Proof of Lemma 3.2.* Using Young's inequality with $p = \frac{1}{\nu}$, $q = p/(p-1) = 1/(1-\nu)$, we have

$$ax^\nu = (cax^\nu) \cdot \left( \frac{1}{c} \right) \leq \frac{1}{p}(cax^\nu)^p + \frac{1}{q}\frac{1}{c^q}.$$

We deduce that $x \leq \alpha x^\nu + b \leq \nu(ca)^{1/\nu} x + (1-\nu)c^{-1/(1-\nu)} + b$.

If we choose $c$ such that $\nu(ca)^{1/\nu} = 1/2$, or equivalently, $c = (2\nu)^{-\nu}a^{-1}$, then

$$\frac{1}{c^{1/(1-\nu)}} = (2\nu)^{\nu/(1-\nu)}a^{1/(1-\nu)}.$$

We deduce that $x \le Ca^{1/(1-\nu)} + 2b$. $\qquad\qquad\qquad\qquad\qquad\qquad\qquad\qquad\qquad$ □

*Proof of Lemma 3.3.* Define $\Gamma(x) = x^2/r - (1-\beta)x + \beta\mathcal{C}$. The minimum value of $\Gamma(x)$ will be attained at $x_0 = r(1-\beta)/2 \le r$ with

$$\Gamma(x_0) = -\frac{r}{4}(1-\beta)^2 + \beta\mathcal{C}.$$

We note that if $\beta < 1 - 2\sqrt{\mathcal{C}/r}$, then $x_0 \ge 1$ and $\Gamma(x_0) < 0$. To ensure $A(l, r, \beta, \mathcal{C}, \alpha) < 0$ for $l = x_0$, we need

$$y = \left[\beta\left(1 - \frac{\alpha}{2}\right) - \frac{1}{2}\right]x_0 + \beta\mathcal{C} = -\frac{r}{4}[1 - \beta(2-\alpha)][1-\beta] + \beta\mathcal{C} < 0.$$

If $\alpha \ge 1$, it is clear that

$$y \le -\frac{r}{4}(1-\beta)^2 + \beta\mathcal{C} = \Gamma(x_0) < 0.$$

If $\alpha \le 1$, we have $y \le -r[1 - \beta(2-\alpha)]^2/4 + \beta\mathcal{C}$; and $y < 0$ if $\beta < (1 - 2\sqrt{\mathcal{C}/r})/(2-\alpha)$. □

*Proof of Lemma 3.6.* We have

$$
\begin{aligned}
\mathbb{E}(\ell(f) - \ell(f^*))^2 &\le \mathbb{E}\ell(f)^2 + \mathbb{E}\ell(f^*)^2 \\
&\le 2\mathbb{E}\ell(f)^2 \le 2[\mathbb{E}\ell(f)^r]^{1/(r-1)}[\mathbb{E}\ell(f)]^{(r-2)/(r-1)} \\
&\le 2W^{r/(r-1)}[\mathbb{E}\ell(f)]^{(r-2)/(r-1)} \\
&\le 2W^{r/(r-1)}\left(\frac{\alpha}{M}\right)^{(r-2)/(r-1)}[\mathbb{E}(\ell(f) - \ell(f^*))]^{(r-2)/(r-1)}.
\end{aligned}
$$

$\qquad\qquad\qquad\qquad\qquad\qquad\qquad\qquad\qquad\qquad\qquad\qquad\qquad\qquad\qquad\qquad$ □

*Proof of Lemma 3.7.* Since $R(f)$ has a unique minimizer at $f^*$, there exists $\alpha > 0$ such that

$$U_\alpha := \{f \in \mathcal{F} : R(f) \le \frac{\alpha}{\alpha - K}R(f^*)\} \subset U.$$

where $M$ is the constant defined in Lemma 3.6. Inside $U_\alpha$, we have

$$\mathbb{E}(\ell(f) - \ell(f^*)) = R(f) - R(f^*) \ge cd^m(f, g) \ge \frac{c}{L^{m/2}}\left(\mathbb{E}(\ell(f) - \ell(f^*))^2\right)^{m/2}. \qquad (0.4)$$

By Lemma 3.6 and (0.4), multi-scale Bernstein's condition holds for $\gamma = ((r-2)/(r-1), 2/m)$. □

*Proof of Corollary 3.1.* Recall that $R(f)$ is strongly convex and $f^*$ is its unique minimizer, we have

$$\frac{R(f) + R(f^*)}{2} \ge R\left(\frac{f+g}{2}\right) + cd^2(f, g) \ge R(f^*) + cd^2(f, g)$$

which implies that

$$\mathbb{E}(\ell(f) - \ell(f^*)) = R(f) - R(f^*) \ge 2cd^2(f, g) \ge \frac{2c}{L}\mathbb{E}(\ell(f) - \ell(f^*))^2.$$

This proves the Bernstein's condition. $\qquad\qquad\qquad\qquad\qquad\qquad\qquad\qquad\qquad\qquad\qquad$ □