[Reviews · NeurIPS 2016]

Reviewer 1

Summary

The paper appears to be the first to give fast learning rates for learning in unbounded domains with heavy tailed loss functions, under conditions introduced. For one of these conditions (multi-scale Bernstein) the authors also discuss how to verify it in practice. The other condition (polynomial entropy bounds on the hypothesis class) is less clear and the discussion section in the end suggests that extensions might be possible. In the case of bounded loss the new result recovers state of the art results of (Mehta & Williamson 2014). The proof uses new results of (Lederer et al 2014) to bound the suprema of empirical unbounded processes. As an application, k-means clustering is analysed in the setting of heavy tailed data distribution.

Qualitative Assessment

This paper appears to be sound and useful. The only question I have is whether the authors could elaborate a bit on how restrictive or not is the polynomial entropy boundedness condition and, if there are indications to "expect" that the result would extend beyond this condition?

Confidence in this Review

2-Confident (read it all; understood it all reasonably well)


Reviewer 2

Summary

Obtaining fast rates of convergence has been a very dynamic field of research these past years, and most of the contributions have been made in the case of bounded loss functions. This paper addresses unbounded loss functions and still derives fast learning rates, at the (mild) price of a multi-scale Bernstein's condition and the existence of the $r$-th moment of the envelope function (upper bound of $\ell\circ \cdot$ where $\ell is the loss function$). The main result (Theorem 3.2) states a sharp oracle inequality holding with high probability, where the rate of convergence is $n^{-\beta}$ and $\beta$ can be made arbitrarily close (from above) to 1 (through $r$: the more moments of the envelope exist, the closer to 1). Section 3.4 is helpful in assessing that the multi-scale Bernstein's condition hold for a particular learning setting. Theorem 4.1 presents an application to this scheme to quantization: the ERM achieves a $\mathcal{O}(n^{-\beta})$ rate where $\beta$ may be arbitrarily close to 1.

Qualitative Assessment

The paper is very pleasant to read and contains stunning contributions. I was impressed by the technical virtuosity and the very dense set of results and ideas contained in just eight pages. The introduction clearly states the paper's ambitions, even for non-expert readers. The main contributions are clearly stated and supported by sound proofs. I believe this work would have an impact on several researchers communities. In my opinion, this is a fine piece of work. Minor point: - Lines 444, 458, 460, 462, 481: please correct to proper citations, where all the authors are credited (no "et al.")

Confidence in this Review

2-Confident (read it all; understood it all reasonably well)


Reviewer 3

Summary

This paper gives learning bounds with heavy-tailed losses in hypothesis classes with logarithmic entropy numbers. If the r-th moment of the supremum loss over all hypotheses in the class is integrable and a 'multiscale Bernstein condition' holds then fast rates ( close to O(n^{-1}) ) can be obtained.

Qualitative Assessment

This paper provides some new results in an important area which is receiving more and more attention: fast rates when loss functions are unbounded and heavy-tailed. Existing results based on empirical process theory often rely on bounded or sub-Gaussian loss, and the heavy tails (hence non-sub-Gaussian) case is considerably harder. The results presented seem sound and are definitely novel. They rely on results of Sara van de Geer and collaborators on concentration inequalities for unbounded empirical processes. The material is very technical and I would suggest moving even some more material to the appendix. This would allow you to include more extensive discussion of some points: The multiscale Bernstein condition is novel and crucial and I would like to see a simple example to explain the difference between the microscopic and macroscopic scales that the authors talk about (the k-means example is nice and really illustrates that the approach is useful, but it didn't really make the intuition of/need for multiscale Bernstein clear to me). In this respect it is of interest that Audibert (Fast Learning Rates..., Annals of Statistics, 2009) shows that a simple , standard Bernstein condition is enough to get fast rates, even for unbounded losses (and it appears even for heavy-tailed losses), if one uses a pseudo-Bayesian estimator with online-to-batch conversion. The authors of the present paper essentially prove uniform convergence rates for the whole class {\cal F}, which implies that e.g. ERM will attain these rates, so it's different from Audibert who uses a very specific randomized estimator and measures error by the expected error according to the predictor's randomization. Nevertheless, it suggests that the multi-scale may perhaps not be necessary, so I would like to some more (informal) explanation of why the authors need it. Do they think it may be dispensed with? (the authors should most definitely check out Audibert's paper. Note that with 'improper learners' that adopt a hypothesis outside the assumed class one can often get better rates, but Audibert 'merely' randomizes estimators). There's also a very recent paper by Mehta and Grunwald (Fast Rates for Unbounded Losses, Arxiv, 2016) where results are proved that are similar in spirit, although the conditions are quite different (yet they seem 'analogous'): instead of the existence-of-moments-of-envelope condition, there is a 'witness condition', and instead of a multiscale Bernstein condition, there is the v-PPC condition (for bounded losses, both multiscale Bernstein and v-PPC conditions are equivalent to the standard Bernstein/central condition; for unbounded losses they seem VERY different). It would be good if the authors could provide some discussion on the relation between these papers (though I concede this does not seem easy; proof techniques are quite different). I would also like to see an explanation of why the approach does not immediately generalize to polynomial entropy numbers. You use chaining in the proofs, so superficially it seems as if the approach would be amenable to larger entropy numbers. You very briefly say in the conclusion that you expect that you can extend it (BTW 'polynomial' there should be 'logarithmic'!) but why isn't the generalization immediate? It should also be pointed out explicitly that the in-probablity results are polynomial and not logarithmic in \delta. This is presumably unavoidable with heavy tails, but it is different from what's normally proved in learning theory - see also the paper by Sabato and Hsu (2015). But let me emphasize that the main results here are quite different from those by Audibert/Grunwald&Mehta/Sabato&Su, and given the challenging nature of this area I do recommend acceptance. Minor issues: The abstract is longer than necessary 'weak-tailed' (beneath Lemma 2.1) is not standard terminology Section 3.4: \gamma-stochastic mixability should be 1-\gamma stochastic mixability (at least that's what I get form Van Erven et al. 2015, I did not check the Mehta&Williamson paper). Thm 4.2, fine -> finite, minimizer -> minimizesrs References: many et al's which should not be there

Confidence in this Review

3-Expert (read the paper in detail, know the area, quite certain of my opinion)


Reviewer 4

Summary

The paper is beyond my expertise for evaluation.

Qualitative Assessment

The paper is beyond my expertise for evaluation. The only interesting part is the application to fast k-means clustering. There is a provable guarantee for faster learning rates. Unfortunately there are no experimental results. It is hard to know the theorem is really useful in practice. Especially, how do we verify the assumption in Theorem 4.1?

Confidence in this Review

1-Less confident (might not have understood significant parts)


Reviewer 5

Summary

This paper studies the fast learning rates when the losses may be unbounded and have a distribution with heavy tails. The authors introduce two new conditions: the first is the multi-scale Bernstein's condition, and the second is the integrability of the envelope function. The learning rate faster than O(n^(-1/2)) is derived.

Qualitative Assessment

A learning rate is obtained for unbounded and heavy-tailed losses by introducing two new conditions. This result extends some exsiting studies and links with some previous conclusions. For clarity, the authors may explain the importance of heavy-tailed losses. An example of heavy-tailed loss would also be helpful. Typo: Assumption 2.1, There exist -> There exists

Confidence in this Review

1-Less confident (might not have understood significant parts)


Reviewer 6

Summary

The article presents an extension of the results of fast learning rates to unbounded losses with a heavy tailed distribution. First, the authors present the initial conditions. Then, they verify these initial conditions. Finally, they show that fast learning rates for heavy-tailed losses can be obtained for the assumed conditions

Qualitative Assessment

In formula, not all used variables are defined (consider general audience). Some variable values are chosen without any explanation or argument. There are few repetations in the text.

Confidence in this Review

1-Less confident (might not have understood significant parts)